environmental chemistry/analytical chemistry/ spectroscopy

graphene oxide/electrospun cellulose nanofibre, adsorption, polar organophosphorus pesticides, electrospinning

**Author for correspondence:**
Sazlinda Kamaruzaman
e-mail: sazlinda@upm.edu.my

# Superhydrophilic graphene oxide/electrospun cellulose nanofibre for efficient adsorption of organophosphorus pesticides from environmental samples

Nor Izzati Fikrah Aris[1], Norizah Abdul Rahman[1], Mohd Haniff Wahid[1], Noorfatimah Yahaya[2,3], Aemi Syazwani Abdul Keyon[4] and Sazlinda Kamaruzaman[1]

[1]Department of Chemistry, Faculty of Science, Universiti Putra Malaysia, 43400 UPM Serdang, Selangor, Malaysia
[2]Integrative Medicine Cluster, Advanced Medical and Dental Institute, Universiti Sains Malaysia, 11800 Pulau Pinang, Malaysia
[3]Department of Chemistry, University of British Columbia, Vancouver, British Columbia, Canada V6T 1Z1
[4]Department of Chemistry, Faculty of Science, Universiti Teknologi Malaysia, UTM Johor Bahru, 81310 11 Johor Bahru, Johor, Malaysia

NY, 0000-0002-3079-7837; SK, 0000-0001-6299-8767

Superhydrophilic graphene oxide/electrospun cellulose nanofibre (GO/CNF) was synthesized, characterized and successfully used in a solid-phase membrane tip adsorption (SPMTA) as an adsorbent towards a simultaneous analysis of polar organophosphorus pesticides (OPPs) in several food and water samples. Separation, determination and quantification were achieved prior to ultra-performance liquid chromatography coupled with ultraviolet detector. The influence of several parameters such as sample pH, adsorption time, adsorbent dosage and initial concentration were investigated. SPMTA was linear in the range of 0.05 and 10 mg l$^{-1}$ under the optimum adsorption conditions (sample pH 12; 5 mg of adsorbent dosage; 15 min of adsorption time) for methyl parathion, ethoprophos, sulfotepp and chlorpyrifos with excellent correlation coefficients of 0.994–0.999. Acceptable precision (RSDs) as achieved for intraday (0.06–5.44%, $n = 3$) and interday (0.17–7.76%, $n = 3$) analyses. Low limits of detection (0.01–0.05 mg l$^{-1}$) and satisfactory consistency in adsorption (71.14–99.95%) were obtained for the spiked OPPs from Sungai

Pahang, Tasik Cheras, cabbages and rice samples. The adsorption data were well followed the second-order kinetic model and fits the Freundlich adsorption model. The newly synthesized GO/CNF showed a great adsorbent potential for OPPs analysis.

## 1. Introduction

Organophosphorus pesticides (OPPs) are among the most broadly used pesticides with high toxicity in the world since the 1970s [1]. Their popularity is largely contributed by their favourable characteristics such as short persistence and biodegradability [2]. The growing usage of organic materials for agricultural purposes such as pesticides causes serious health risks to the animal and human. Methyl parathion (*O,O*-dimethyl *O-p*-nitrophenylphosphorothioate), sulfotepp (diethoxyphosphinothioyloxy-diethoxy-sulfanylidene-$\lambda^5$-phosphane), ethoprophos (1-(ethoxy-propylsulfanylphosphoryl)sulfanylpropane) and chlorpyrifos (*O,O*-diethyl *O*-3,5,6-trichloropyridin-2-yl phosphorothioate) are examples from the most used classes of OPPs. Over so many years, pesticides have been used by growers and farmers for food production in order to increase the quality and the production of crops and against a wide range of pests. Due to sparing of plants or crops, only some pesticides remain in crops in significant amounts and others leach out in water, damaging the environment, and disperse into the environment through wind, rain and fog [3,4]. As a consequence, consumers are exposed to pesticides even though the exposure is usually in a small number of quantities [5].

OPPs may have effects on untargeted organisms including humans upon a frequent exposure for a long period of time to even low (ppb) levels of OPPs from air, water and food due to the acetylcholinesterase inhibition in the body and are reported to exhibit teratogenicity, cytotoxicity, immunotoxicity and genotoxicity [2,6]. Thus, this results in dysfunction of many behavioural and autonomic systems, eventually leading to muscle and respiratory weakness or paralysis as well as death [1,7]. Analysis and monitoring of trace levels of OPPs in environmental and food contamination are very crucial, as the maximum residue limits (MRLs) set by the European Union for pesticide residues in vegetables and fruits is 0.01 mg kg$^{-1}$ [5] and a maximum level for total pesticides that are intended for human consumption in water is 0.5 µg l$^{-1}$ [8]. Contamination of agricultural products and environmental water has become a major concern, as OPPs are one of the most common classes of pesticides involved in poisoning. Stability and difficult degradation of OPPs are due to their complex aromatic structure. Thus, analysis and monitoring of trace levels of OPPs in environmental water and food contamination are very crucial for environmental control [5].

OPPs determination constitutes a real analytical challenge as they are recognizable in very low concentration levels of environmental water and food samples. These OPPs have the possibility to be analysed by liquid chromatography (LC) as they exhibit a wide range of physico-chemical properties. Ultra-performance liquid chromatography (UPLC) is a chromatographic technique that has overcome conventional high-performance liquid chromatography (HPLC) limitations such as the speed of analysis and lower separation capacity [9].

Since the OPPs concentration in the real samples are in trace amounts, various techniques have been developed like photocatalysis, oxidation [7], coagulation and flocculation [4], ion-exchange [4] and adsorption [10] for the adsorption of these pesticides. Adsorption of these pesticides by using the oxidation method is difficult as this method has significant limitations, such as the massive production of iron sludge, and high-cost requirements [11]. Coagulation and flocculation of pesticides produces large amounts of sludge at their disposal which becomes a problem [12]. Adsorption has recently been considered as a capable, acceptable and most important technique for decreasing concentration of OPPs from aqueous solutions due to its availability of different adsorbents, simple, economical, low-cost and environment-friendly. Previously, the universal method used for OPPs analysis in food and water is solid-phase adsorption [13]. However, this method has several drawbacks, such as it requires larger quantities of organic solvents and adsorbent, which limits the extensive deployment of solid-phase adsorption [14]. Pipette tip solid-phase adsorption is an attractive alternative to normal solid-phase adsorption in terms of the set-up. This miniaturized method offers the superior and unique combination of the shorter operation period, lower analysis costs, lower consumption of organic solvent and adsorbent dosage, and economic methods of targeting multiple analytes [15,16]. This method involves the utilization of tiny adsorbent held in a holder as an adsorption medium, and it relies on the fact that most of the analytes are adsorbed in the adsorption region of the adsorbent through some interaction by stirring the solution. This adsorption corresponds specifically to the bonds present in the analytes and the sorbent. In general, the efficiency of pipette tip solid-phase adsorption relies on the adsorbents performance. Thus, an adsorbents exploration with great adsorption capacity is essential [17].

Recently, efforts have been made to use more affordable adsorbent in the adsorption of pesticides. Adsorbents that have been reported for fast adsorption of pesticides are organic porous polymers [4], cellulose acetate (CA) [18], silica particles [7], graphene oxide (GO) [19] and multi-walled carbon nanotubes [20]. CA has been widely investigated as a type of hybrid adsorbent as compared with the other materials and exhibits potential applications for pesticides adsorption. Various studies have been done by using CA as an adsorbent in an adsorption of different contaminants such as separation/preconcentration of silver(I) and lead(II) in environmental samples on cellulose nitrate membrane filter prior to their flame atomic absorption spectrometric determinations [21], cellulose cellets as a new type of adsorbent for the removal of dyes from aqueous media [22] and preparation of cellulose/graphene composite and its applications for triazine pesticides adsorption from water [18].

CA is highly hydrophilic in nature as it consists of high density of carbonyl and hydroxyl groups and has high strength, excellent stability, large specific surface area and might be moulded easily into different forms, such as spheres, fibres and membranes, making them a suitable candidate as a mechanical support and an adsorbent [16,23]. CA has strong inter- and intra-molecular hydrogen bondings that are responsible for the hydrophilic nature of the polymer. However, CA has some limitations in terms of high moisture absorption, associated dimensional instability and ultraviolet (UV) stability. CA also can be easily attacked by concentrated alkalies and acids. Therefore, CA could be tailored and modified with GO in order to overcome the limitation and enhance the OPPs adsorption capacity [23]. Recently, GO, which is a class of two-dimensional material, has been broadly used in many research fields for their high specific surface area and high adsorption capacity. Moreover, GO has a high mechanical strength range of oxygen functional groups, for instance, ketone, epoxides, carbonyls, carboxylic and hydroxyl group on both edges and basal planes [24,25], thus making GO a hydrophilic material. These oxygen functionalities provide the affinity towards a great number of compounds over different interactions, such as $\pi$–$\pi$ stacking dispersion forces, hydrogen bonding and electrostatic interaction [1], which are ideal characteristics for an adsorbent as it extends the application and enhances the material properties towards OPPs [26].

Combining CA and GO could be a feasible way to overcome the limitations for the adsorption of polar OPPs from food and water samples. Introduction of GO onto CA will give a narrow pore-size distribution and improve their adsorption capacity. In enhancing the capabilities and advantages of this adsorbent, a special technique, electrospinning, is introduced. The main advantage of the electrospinning process is the ability to form a fibrous membrane with excellent pores interconnection and expanded functional groups as well as high porosity [27]. The combination of CA and GO is considered to enhance the efficiency, strengthen the structure and reinforce the membrane as the addition of inorganic or organic materials to electrospun solution leads to the production of nanofibres with additional properties. So, in this work, graphene oxide/electrospun cellulose nanofibre (GO/CNF) was successfully synthesized, characterized and used for the adsorption of polar pesticides, namely methyl parathion, ethoprophos, sulfotepp and chlorpyrifos, from food and water samples prior to ultra-performance liquid chromatography coupled with ultraviolet (UPLC–UV) determination. The effect of initial pH, contact time, adsorbent dosage, initial concentration of OPPs and desorption solvent was evaluated. In addition, adsorption kinetics and isotherm study were also investigated and analysed. To the best of our knowledge, no report has been published to discuss in detail: (i) on the adsorption study and (ii) on the validated adsorption method towards polar OPPs by using superhydrophilic GO/CNF as an adsorbent.

# 2. Material and methods

## 2.1. Chemicals and reagents

The standards used were of the highest purity available (greater than or equal to 98%). Methyl parathion, ethoprophos, sulfotepp and chlorpyrifos were purchased from Sigma–Aldrich (Schnelldorf, Germany). Analytical grade reagents/chemicals were used. CA was obtained from Sigma–Aldrich (Pty.). Hydrochloric acid (37%), HPLC grade acetonitrile (99.9%), sodium hydroxide, *N,N*-dimethylformamide (DMF) and graphene oxide were from QReC (Selangor, Malaysia).

## 2.2. Instrument

GO/CNF was synthesized using an electrospinning machine through a syringe pump (NE-300 Just Infusion™ Syringe Pump), which was connected through a high-voltage power supply (RXN-305D, China) at 2 ml h$^{-1}$ of flow rate by an 18-gauge needle. Fourier transform infrared (FTIR) spectra (4000–500 cm$^{-1}$) were

recorded on a Perkin Elmer (1600 series) spectrometer (Waltham, MA, USA) using an attenuated total reflectance (ATR) technique. The morphology, structure and diameter of the new GO/CNF were observed using a JEOL JSM-6700F field emission scanning electron microscopy (FESEM; Tokyo, Japan) operated at 5.0 kV, magnification 10 000× and a working distance of 4.3 mm. OPPs analysis were carried out using UPLC through a Supelco Ascentis® C18 column (15 cm × 4.6 mm, 5 µm) (Waters, Ireland).

## 2.3. UPLC with UV detector analysis

UPLC analysis was carried out on a Supelco Ascentis® C18 column (15 cm × 4.6 mm, 5 µm) (Waters, Ireland). Isocratic elution was carried out by the mobile phase of acetonitrile–0.1% acetic acid (70 : 30, v/v) with a flow rate of 0.2 ml min$^{-1}$. The detection of analytes was carried out with the UV detector at 280, 230, 220 and 200 nm for each analyte (chlorpyrifos, sulfotepp, ethoprophos and methyl parathion), respectively, with 2 µl of the injection volume. In isocratic mode, the injection volume and the mobile phase flow rate are the important parameters that need to be modified. They are the mobile phase flow rate and the injection volume. In order to maintain equivalent sensitivity and to avoid a detrimental extra column band broadening, it is necessary to adapt the injection volume in line with the change of column dimensions. In LC, the injected volume should represent only 1–5% of the column volume. Therefore, the injection volume is independent of the particle size and only proportional to the column volume.

## 2.4. Standard solution and sample preparation

The stock solutions (1000 mg l$^{-1}$) of methyl parathion, ethoprophos, sulfotepp and chlorpyrifos were stored and secured from light at 4°C after being separately prepared in acetonitrile. A few proper dilutions of the stock solution for each analyte were prepared at 50 mg l$^{-1}$ and 10 mg l$^{-1}$. The water samples were freshly prepared with spiking analytes at 0.1 mg l$^{-1}$, and the required working solutions were prepared by a proper dilution of the stock solutions using distilled water.

Tasik Cheras and Sungai Pahang were filtered using a nylon filter paper (0.2 µm) for the removal of any debris. Chopped cabbages and rice were homogenized in a blender with microcutters. Ten grams of each sample was homogenized with 20 ml (1 : 1) of water and acetonitrile by sonication for 30 min. The mixture was filtered using a Buchner funnel and the solid residue was washed through 10 ml of distilled water. The filtrate was used as a real sample and the sample solution was spiked at the desired concentrations with the four analytes for further analysis. Two different concentrations of analytes in water (0.5 and 5 mg l$^{-1}$) and food samples (0.1 and 0.5 mg l$^{-1}$) were prepared, respectively.

## 2.5. Superhydrophilic GO/CNF preparation and characterization

Electrospinning was conducted on an electrospinning apparatus at room temperature. CA (1 g) was dissolved in a 15 ml mixture of acetone/DMF (2 : 1). The CA solution (15 ml) was then mixed thoroughly with different amounts of GO solution (1, 3, 5, 7 and 10 wt%). The amounts of GO used for each wt% were 0.15, 0.45, 0.75, 1.05 and 1.5 ml, respectively. Homogeneous CA solution and 1–10 wt% GO mixed solution was pumped through a syringe at 2 ml h$^{-1}$ of constant flow rate by an 18-gauge stainless steel needle using a syringe pump (NE-300 Just Infusion™ Syringe Pump), which was connected through a high-voltage power supply (RXN-305D, China). An aluminium foil was used as the collector. During the process, the distance from the syringe tip to the collector and the electric potential were fixed, respectively, at 15 cm and 20 kV. The collected GO/CNF on an aluminium foil was stored in a dry place and ready to be used for OPPs adsorption as an adsorbent. The real photograph of the nanofibre is shown in figure 1a.

FTIR spectroscopy was carried out on a Perkin Elmer (1600 series) spectrometer. A solid sample on the universal diamond top-plate, the ATR was pressurized to produce spectra. The spectra were recorded in the range of 500–4000 cm$^{-1}$. The morphology of the GO/CNF was observed by using a JEOL JSM-6700F FESEM. A small piece of the sample was coated with an extremely thin layer (1.5–3.0 nm) of gold-palladium or gold to achieve conductivity and vacuum durability. After being covered by a conductive layer, the sample was mounted on a special holder and scanned with an electron beam. Pure CA and several CA with GO adsorbent at the following loading of 1, 3, 5, 7 and 10 wt% were analysed using FESEM. FESEM is the most important analysis in determining the best adsorbent that will be used throughout the adsorption study through the determination of adsorbent diameter and the formation of GO beads. BELSorp Mini II was used to determine the pore volume and the surface area of the GO/CNF. For the measurement, the adsorbent was outgassed at 350°C under nitrogen flow for at least 3 h and the nitrogen adsorption–desorption data were recorded at liquid nitrogen temperature (77.3 K).

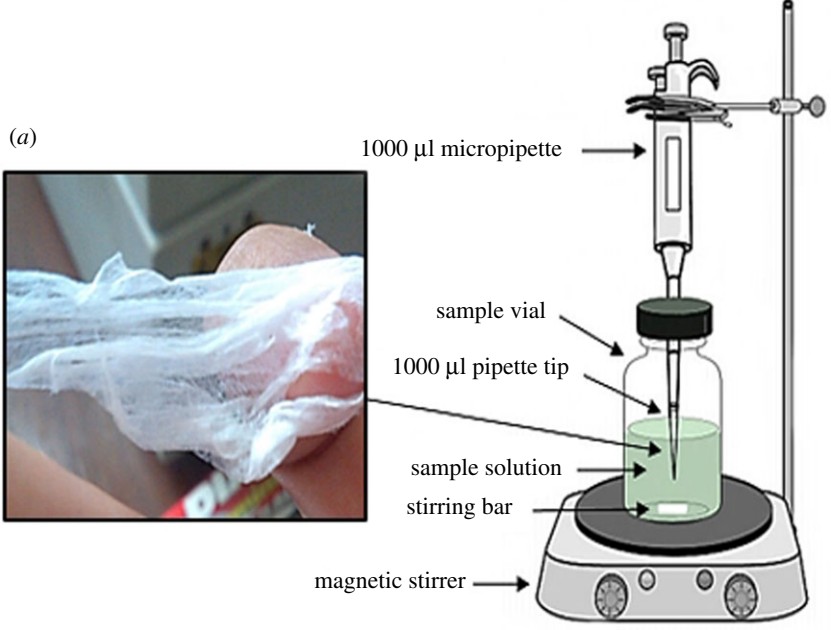

**Figure 1.** The schematic of SPMTA set-up and (*a*) the real photograph of GO/CNF.

## 2.6. Adsorption study

The solid-phase membrane tip adsorption (SPMTA) device consisted of superhydrophilic GO/CNF attached to 1000 µl capacity pipette tip. The procedure was carried out as described previously [28] prior to some modifications. A 7 mm length of 1000 µl pipette tip-end was cut off. Then, prior to adsorption, the GO/CNF–10 wt% was inserted fully fitted into the tip. The SPMTA device was exposed to the sample solution during the adsorption as shown in figure 1. The tip was placed into the sample solution that was stirred continuously for 15 min at 1000 r.p.m. Then, 1000 µl of sample solution was withdrawn manually into the tip using a digital micropipette (Gilson, USA) for every 5 min interval where a dynamic adsorption was performed. The withdrawn sample was released back into the sample vial after a dwelling time of roughly 10 s with the same speed. After adsorption, the tip was detached and 2 µl from the sample solution was injected into the UPLC–UV. Figure 1 shows the schematic of SPMTA set-up and the real photograph of GO/CNF. Each parameter was optimized accordingly in a sequence, pH, time, adsorbent dosage, initial concentration and desorption study. This was followed by the performance of the method and application to the real samples.

## 2.7. Adsorption capacity of OPPs onto GO/CNF

The total capacity of analytes adsorbed by the GO/CNF is denoted by $q_e$ (adsorption capacity). The $q_e$ was intended by the final and initial analytes ion concentration differences using the following equation [29]:

$$q_e = (C_i - C_f) \times \frac{V}{m},\qquad(2.1)$$

where $q_e$ represents the adsorption capacity (mg g$^{-1}$); $C_f$ and $C_i$ are, respectively, the final and initial concentration of the solution (mg l$^{-1}$); $V$ is denoted as the volume of the solution (l), and $m$ represents the mass of an adsorbent (g).

## 2.8. Kinetic and isotherm study

In the kinetics of the adsorption process, the pseudo-first order and the pseudo-second order model were determined. The linear form of the pseudo-first order model is expressed as follows:

$$\ln(q_e - q_t) = \ln(q_e) - k_1 t,\qquad(2.2)$$

where $q_e$ and $q_t$ are denoted as the adsorption capacity, respectively, at equilibrium and at time $t$ (mg g$^{-1}$), and $k_1$ represents the adsorption rate constant (min$^{-1}$). The pseudo-second order model is presented in a linear form as follows:

$$\frac{t}{q_t} = \frac{t}{q_e} + \frac{1}{k_2 q_e^2},$$  (2.3)

where $q_e$ and $q_t$ represent the adsorption capacity, respectively, at equilibrium and at time $t$ (mg g$^{-1}$), and $k_2$ denoted as the rate constant (g mg min$^{-1}$).

Isotherm study was given by the Langmuir and Freundlich isotherm. The Langmuir isotherm is presented in a linear form as follows:

$$\frac{C_e}{q_e} = \frac{1}{K_L q_m} + \frac{C_e}{q_m},$$  (2.4)

where $q_e$ and $q_m$ represent the adsorption capacity and monolayer of the adsorbent capacity (mg g$^{-1}$), respectively, $C_e$ represents the equilibrium concentration (mg l$^{-1}$), and $K_L$ is the Langmuir constant (l mg$^{-1}$).

The Freundlich model is given in a linear form as follows:

$$\ln(q_e) = \ln(K_F) + \frac{1}{n} \ln(C_e),$$  (2.5)

where $q_e$ is the adsorption capacity (mg g$^{-1}$), $C_e$ represents the equilibrium concentration (mg l$^{-1}$), and $n$ and $K_F$ are the Freundlich constants (l mg$^{-1}$).

## 2.9. Desorption study

The reusability of GO/CNF loaded with OPPs analytes can be achieved by using desorption and regeneration study using different desorption solvents (1 M hydrochloric acid, methanol, toluene, dichloromethane and isopropyl alcohol). The amount of analytes desorbed from the adsorbent is represented by the percentage of desorption and is expressed as follows:

$$\text{desorption}\ (\%) = \frac{C_{des}}{C_{ad}} \times 100,$$  (2.6)

where $C_{des}$ and $C_{ad}$ (mg l$^{-1}$) are, respectively, the desorbed and adsorbed concentration of the analytes.

## 2.10. Performance of the method

Analytical method performance is necessary as to ensure that every measurement in the routine analysis will be close enough to the unknown true value for the content of the analytes in the samples. Since the sensitivity of the developed method is very important, linearity, consistency in adsorption, repeatability, limit of detection (LOD) and limit of quantification (LOQ) were determined. The linearity was measured in the range of concentration of 0.05–10 mg l$^{-1}$. A consistency in adsorption was achieved when the SPMTA technique was applied for the adsorption of OPPs from water and food samples. The consistency in adsorption and RSDs were determined with OPPs spiked at two different concentrations, 0.5 and 5 mg l$^{-1}$ in water samples and 0.1 and 0.5 mg l$^{-1}$ for food samples, respectively. The LOD and LOQ were determined based on the standard deviation and slope of the calibration curve ($\sigma$/S) method with the ratio of 3 and 10, respectively ($3 \times (\sigma/S)$ and $10 \times (\sigma/S)$). The consistency in adsorption was calculated based on the percentage of the difference in mean values of adsorbed analytes and the analyte concentration against the concentration of analyte. The percentage of consistency in adsorption and method precision (RSD) were calculated as follows:

$$\text{consistency in adsorption}\ (\%) = \frac{(\bar{x} - x)}{x} \times 100$$  (2.7)

and

$$\text{RSD}\ (\%) = \frac{x - \bar{x}}{\bar{x}} \times 100,$$  (2.8)

where $\bar{x}$ and $x$ (mg l$^{-1}$) represent the mean value and concentration of the target analytes.

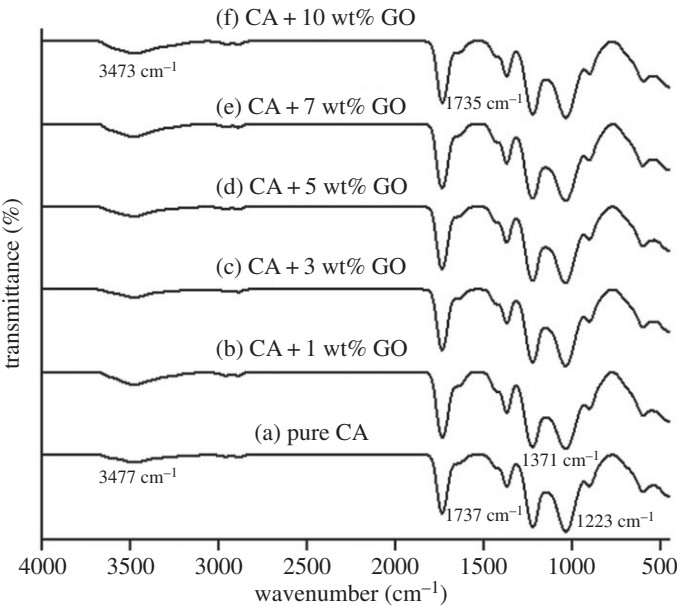

**Figure 2.** FTIR spectra of pure CA (a) and GO-incorporated CA hybrid nanofibres with (b) 1 wt% of GO, (c) 3 wt% of GO, (d) 5 wt% of GO, (e) 7 wt% of GO and (f) 10 wt% of GO.

## 2.11. Real water and food samples study

For this study, selected environmental water samples (Sungai Pahang and Tasik Cheras) and food samples (cabbages and rice) were tested. The optimum conditions adapted from all parameters (sample pH 12; 5 mg of adsorbent dosage; 15 min of adsorption time) were applied in determining the consistency in adsorption (%) at two different concentrations of OPPs spiked in water (0.5 and 5 mg l$^{-1}$) and food samples (0.1 and 0.5 mg l$^{-1}$), respectively.

# 3. Results and discussion

## 3.1. Superhydrophilic GO/CNF characterization

The interaction between GO and CA is illustrated in figure 2. The FTIR spectral studies were applied on the pure CA as well as CA with GO at the loading of 1, 3, 5, 7 and 10 wt%. FTIR spectrum of pure CA (figure 2a) shows characteristic peaks at 1737, 1371 and 1223 cm$^{-1}$, respectively, due to the C–O–C, C=O and C–CH$_3$ stretching. The peak around 1737 cm$^{-1}$ is due to the carbonyl vibration in GO (–COOH groups) or in the CA (–COOR groups). After modification with GO, there is a little shift in peak formed from 1737 to 1735 cm$^{-1}$ (figure 2a–f) with the increase in GO content. It means that the state of C=O (carbonyl groups) has altered from –COOR to partially –COOH that is in GO. The broad adsorption around 3477 cm$^{-1}$ is due to the –OH groups (figure 2a). As GO content increases, this peak becomes stronger since extra –OH groups are on the fibre. This peak shows a shift from 3477 to 3473 cm$^{-1}$ (figure 2a–f). The shift in the absorption band strongly suggests strong interactions between –OH groups of GO and C=O groups of CA through hydrogen bonding. So, the differences in the spectrum patterns show the success of modified GO/CNF from CA. In this study, FESEM analyses enable the direct observation of the surface microstructures of electrospun CA and GO/CNF. The FESEM images (figure 3) show the variation of the adsorbent diameters triggered with different GO loading. This distinction shows that the largest diameter was on pure CA and the diameters decrease with the increasing weight of GO. The diameter of pure CA nanofibres (figure 3a) is approximately 0.33 µm and the diameter of GO/CNF composite (figure 3b) decreased to be 0.21 µm. Figure 3c and d (close-up images) shows the average adsorbent diameter of about 0.18 and 0.17 µm, respectively. In addition, 0.15 and 0.11 µm are approximately the average adsorbent diameter shown in figure 3e and f (close-up images), respectively. The fibre diameter was affected by some parameters during the electrospinning process such as the power supply volt and the pump flow rate. In this entire procedure, 2 ml h$^{-1}$ and 20 kV were used and fixed. As GO content increases, the surface

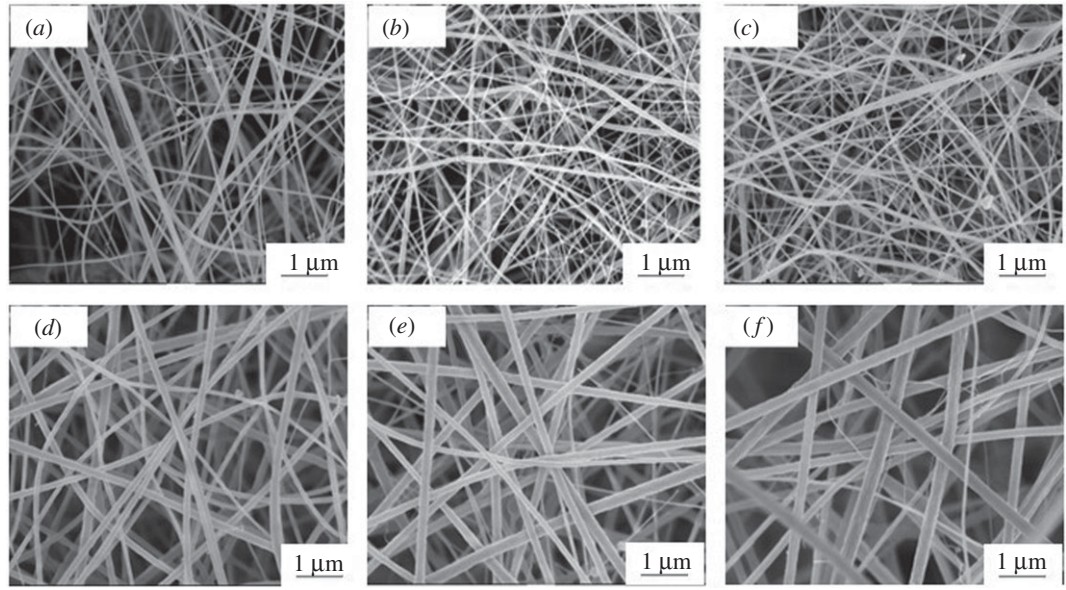

**Figure 3.** FESEM micrographs of (*a*) pure CA, (*b*) GO/CNF–1 wt%, (*c*) GO/CNF–3 wt%, (*d*) GO/CNF–5 wt%, (*e*) GO/CNF–7 wt% and (*f*) GO/CNF–10 wt%. Close-up images for *d*, *e* and *f*.

**Table 1.** Preliminary adsorption of OPPs using various adsorbents.

|  | adsorption capacity ($q_e$) | | | |
| --- | --- | --- | --- | --- |
|  | methyl parathion | ethoprophos | sulfotepp | chlorpyrifos |
| pure CA | 0.031 | 0.010 | 0.011 | 0.010 |
| GO/CNF–1 wt% | 0.030 | 0.021 | 0.010 | 0.031 |
| GO/CNF–3 wt% | 0.033 | 0.040 | 0.032 | 0.030 |
| GO/CNF–5 wt% | 0.030 | 0.025 | 0.034 | 0.038 |
| GO/CNF–7 wt% | 0.036 | 0.132 | 0.040 | 0.037 |
| GO/CNF–10 wt% | 0.039 | 0.221 | 0.075 | 0.069 |

tension of the solution increases [30]. Therefore, to defeat the surface tension of the solutions, the use of higher electric field and power supply voltage during the electrospinning process are necessary [31]. The diameter of the fibres decreases as a result of a higher rate of electrostatic stretching [32]. The flow rate needs to be reduced as GO content increases so as to decrease the adsorbent diameter [31]. As a result, it confirmed that at higher GO contents, the formation of continuous hybrid nanofibres was forbidden due to incapability to preserve the stable flow of the polymeric solution at the tip of the needle [32]. The formation of GO beads at the adsorbent may suggest that the GO was not evenly dispersed in the CA solution at such higher concentration. That is why the analysis of the GO/CNF was only up to 10 wt% of GO loading so as to preserve the stable flow and to avoid the hardening of the solution at the tip of the needle as well as the prevention of the formation of GO beads. The formation of the electrospun nanofibre using higher than 10 wt% of GO loading was impossible as the formation of the polymer jets was forbidden due to the incapability to form a polymer droplet. Preliminary adsorption of OPPs analytes using different percentages of GO adsorbent was performed and analysed. The adsorption capacity ($q_e$) achieved for each adsorbent in methyl parathion, ethoprophos, sulfotepp and chlorpyrifos are presented in table 1. From the achieved results, the GO/CNF was successfully synthesized and 10 wt% GO shows the highest adsorption capacity throughout all the analytes (0.039, 0.221, 0.075 and 0.069 mg g$^{-1}$). The GO/CNF–10 wt% is believed to be the best adsorbent among the rest and will be applied in SPMTA as an adsorbent for subsequent analysis.

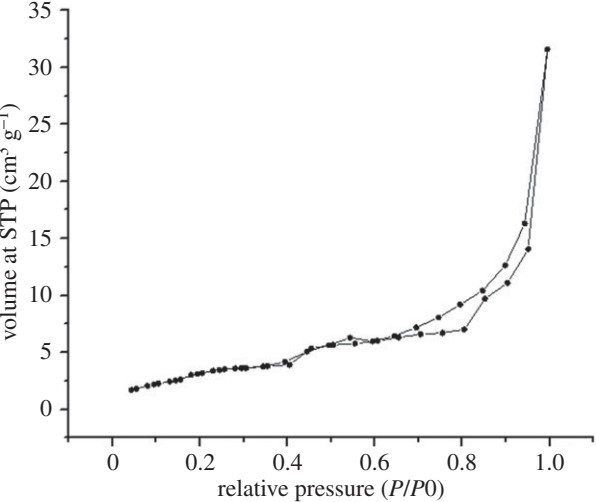

**Figure 4.** Nitrogen adsorption–desorption isotherm of superhydrophilic GO/CNF–10 wt%.

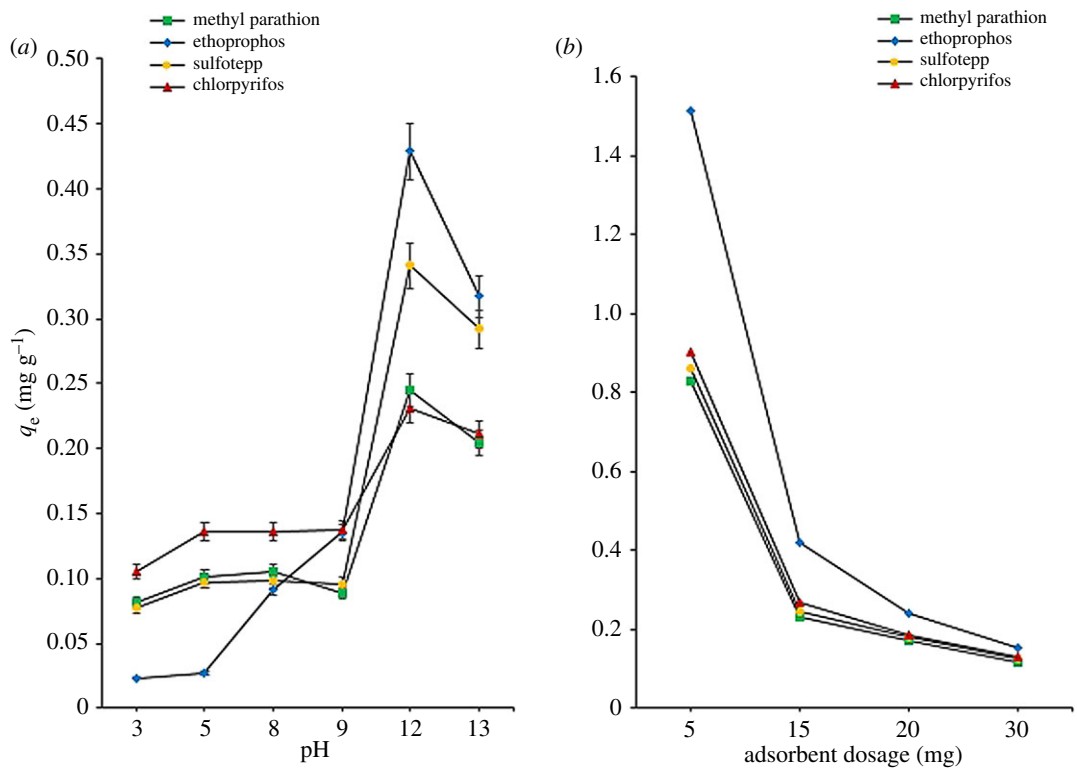

**Figure 5.** Effect of (*a*) sample pH and (*b*) adsorbent dosage on adsorption study of OPPs. Adsorption conditions: 0.1 mg l$^{-1}$ of spiked OPPs in 10 ml sample solution; adsorption time 20 min (error bars represent standard deviations of results, $n = 3$).

BET analysis is the most common method of surface area determination and indicates the physical adsorption of molecules on an adsorbent surface. The shape of the adsorption isotherm provides qualitative information on the adsorption process (figure 4). The adsorbent is supposed to possess certain porosity for a specific type of isotherm. The obtained specific surface area of the GO/CNF– 10 wt% from BET was 3.391 m$^2$ g$^{-1}$. Electrospun CA nanofibres were reported to exhibit a surface area of 2.9 m$^2$ g$^{-1}$ [33]. Therefore, the electrospun nanofibre in this research work exhibited a larger BET surface area. In conjunction with a higher surface area, it can be concluded that the GO/CNF–10 wt% has a higher potential in the adsorption, which is a desirable characteristic in high-efficiency adsorption material. Surface area is one of the very attractive attributes of adsorbent for separation technique. Thus, the higher the surface area, the higher the quantity of analytes that can be adsorbed

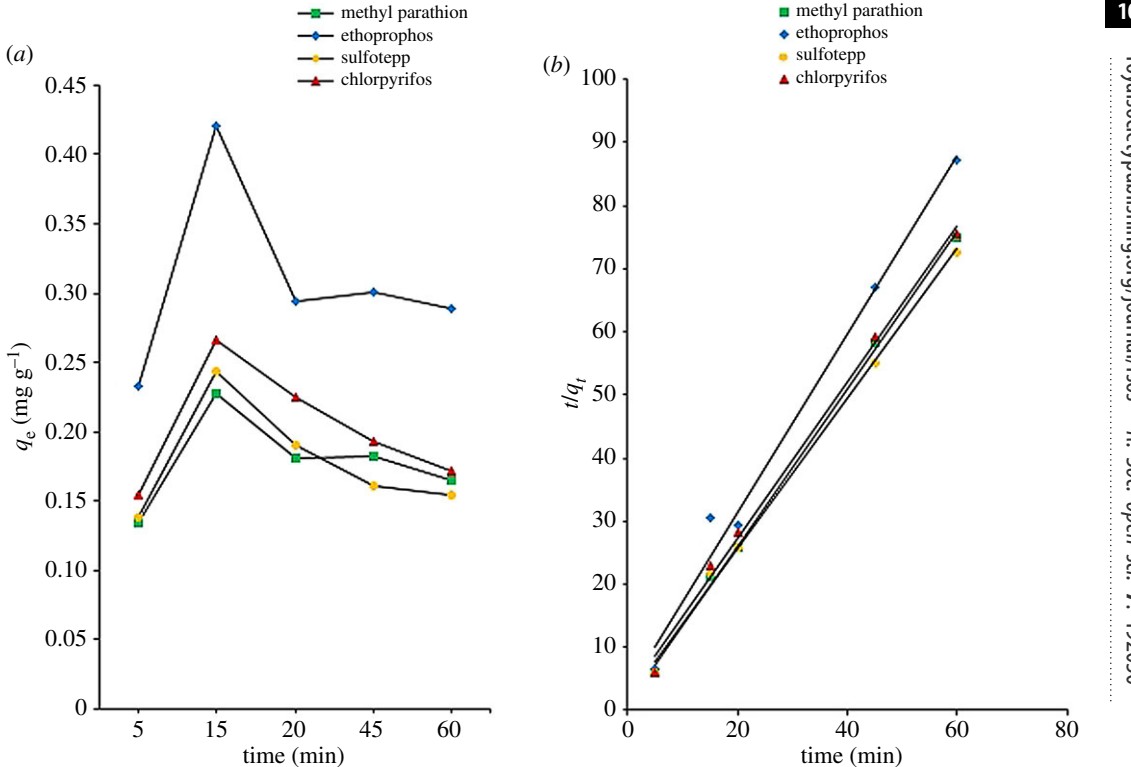

**Figure 6.** (a) Effect of contact time on adsorption study of OPPs and (b) pseudo-second order kinetic model for the adsorption of OPPs onto electrospun GO/CNF–10 wt%. Adsorption conditions: 0.1 mg l$^{-1}$ of spiked OPPs in 10 ml sample solution; sample pH 12; 5 mg of GO/CNF–10 wt%.

on the adsorbent surface. Hence, it is expected to be efficient to adsorb analytes from environmental water and food samples.

## 3.2. Adsorption study

SPMTA parameters were optimized for sample pH, adsorbent dosage, adsorption time and initial concentration. These parameters were optimized one at a time keeping the other parameters constant in three adsorption determinations ($n = 3$) for each analysis using 0.1 mg l$^{-1}$ of spiked OPPs in 10 ml sample solution. Initially, 15 mg of adsorbent and 20 min of adsorption time were used.

## 3.3. Sample pH

The distribution of the negative and positive charges may have huge effects on an adsorption by the adsorbent. Thus, the balance between them is one of the most important environmental factors that affect the adsorbent potential [34]. In this study, the effect of pH was examined between 3 and 13, and the results are presented in figure 5a. From the graph, the highest adsorption was achieved at pH 12 with adsorption capacity of 0.25, 0.43, 0.34 and 0.23 mg g$^{-1}$ for methyl parathion, ethoprophos, sulfotepp and chlorpyrifos, respectively. Ethoprophos shows the highest adsorption among all analytes, while chlorpyrifos was the worst. Ethoprophos has higher donating electron ability of S, N and O atoms as compared to other analytes. Two S atoms are present in the structure, which assisted with the higher adsorption.

The adsorption process occurred more favourably when all analytes were in the neutral form. There are many possible interactions which may occur during the adsorption process between the GO/CNF–10 wt% adsorbent and the OPPs such as π–π interaction, electrostatic interactions and hydrogen bonding. The decreasing adsorption capacity when pH was adjusted to be less than pH 12 might be due to the protonation and deprotonation of OPPs compound and led to less interaction towards the adsorbent. Analytes adsorption is strongly dependent on the pH values. The increase of pH value leads to a much higher adsorption due to the increase in the negative adsorption sites available to bind and a higher adsorption capacity. In a strong acidic medium (pH 3), the adsorption capacity was affected

**Table 2.** Pseudo-first order and pseudo-second order parameters for OPPs adsorption onto GO/CNF–10 wt%.

| | | pseudo-first order | | | pseudo-second order | | |
|---|---|---|---|---|---|---|---|
| | experimental $q_e$ (mg g$^{-1}$) | $k_1$ (min$^{-1}$) | calculated $q_e$ (mg g$^{-1}$) | $R^2$ | $k_2$ (g mg min$^{-1}$) | calculated $q_e$ (mg g$^{-1}$) | $R^2$ |
| methyl parathion | 0.23 | 0.0017 | 0.67 | 0.235 | 1.53 | 0.80 | 0.998 |
| ethoprophos | 0.42 | 0.0056 | 0.50 | 0.603 | 0.64 | 0.71 | 0.986 |
| sulfotepp | 0.24 | 0.0005 | 0.68 | 0.045 | 0.78 | 0.84 | 0.997 |
| chlorpyrifos | 0.27 | 0.0018 | 0.66 | 0.414 | 0.64 | 0.81 | 0.996 |

due to the presence of excess hydrogen ion and significant competition between the hydroxyl ion groups of adsorbent as well as surrounded hydrogen ions on pesticides for adsorption sites. Thus, at lower pH values, the adsorption occurred slightly and undergoes repulsion between the sorbate to the sorbent as the hydrogen ions strongly compete with OPPs for the same adsorption sites on the adsorbent surface [35]. As the solution pH value increases, the adsorption of OPPs develops significantly and maximizes at pH 12. This happened as the concentration of the hydrogen ions as competitor decreased as the adsorbent surface became more negatively charged. A gradual adsorption capacity decrease observed as sample pH was further increased to 13. At this point, the analytes started to react with hydroxide ions to form OPPs hydroxide (conjugated form) and started to be precipitated (ionized) [19]. Therefore, the opportunity of OPPs analytes to bind to the adsorbent surface is lessened [19]. Other than that, at this point, OPPs analytes undergo hydrolysis rather than adsorption [19]. Hydrolysis is due to the reaction with water. At higher pH, water reacts with an alkaline solution and the analyte compounds itself broken down by the alkaline environment. Thus making a significant contribution to the OPPs analytes removal. Thus, pH 12 was selected for the sample solution prior to the subsequent investigation.

## 3.4. Contact time and adsorption kinetic

Kinetic studies in which the relationship between the contact time and adsorption capacity are among the most important studies related to adsorption. The speed of the adsorption process can be predicted and the data from these predictions can be used for the modelling and designing of an adsorption process [34]. The adsorption time was optimized within the range of 5–60 min (figure 6a). The adsorption for all analytes increased within 15 min, which was possibly owing to the ample availability of adsorbent's active sites and also due to the rapid attachment to the adsorbent surface [18]. The active sites had a tendency to saturate with the increase of contact time. Beyond 15 min, the adsorption efficiency for all the analytes decreased as this might probably be due to the inter-analyte competition for adsorbent sites and displacement, and also analytes back-adsorption from the adsorbent into the sample solution [36]. At 15 min, the adsorption capacities of GO/CNF–10 wt% for 0.1 mg l$^{-1}$ of methyl parathion, ethoprophos, sulfotepp and chlorpyrifos were 0.228, 0.421, 0.243 and 0.258 mg g$^{-1}$, respectively. Methyl parathion shows the lowest adsorption capacity, as an additional branch in the structure weakens the π-bonding network of benzene in its structure and the electrostatic type interaction. This branched structure is not conducive to the interactions between the methyl parathion and the fibre itself. As a consequent, 15 min was chosen for further experiments.

Pseudo-first order and pseudo-second order adsorption kinetic models explain the adsorption mechanisms and process as these models were used to describe the adsorption efficiency. The data of the adsorption capacity at time $t$ and at equilibrium ($q_t$ and $q_e$) were fitted with the two models. Commonly, adsorption is a process of a complex multistep and these kinetic studies will provide valuable insights of mechanisms of the adsorption. All parameters of kinetics (rate constants ($k_1$ and $k_2$) and correlation coefficients, $R^2$) are summarized in table 2.

Higher correlation coefficients ($R^2 \geq 0.986$) were shown in the pseudo-second order model (figure 6b) as compared with the pseudo-first order model (not shown) ($R^2 \geq 0.045$). A nonlinear fitting presented in the pseudo-first order model as the rate constant ($k_1$) depends on the initial analytes concentration and varies significantly according to the adsorption system [37]. The rate constant, $k$, gives a direct measure of the relative reaction rate. Pseudo-second order rate constant ($k_2$) has a higher value compared to pseudo-

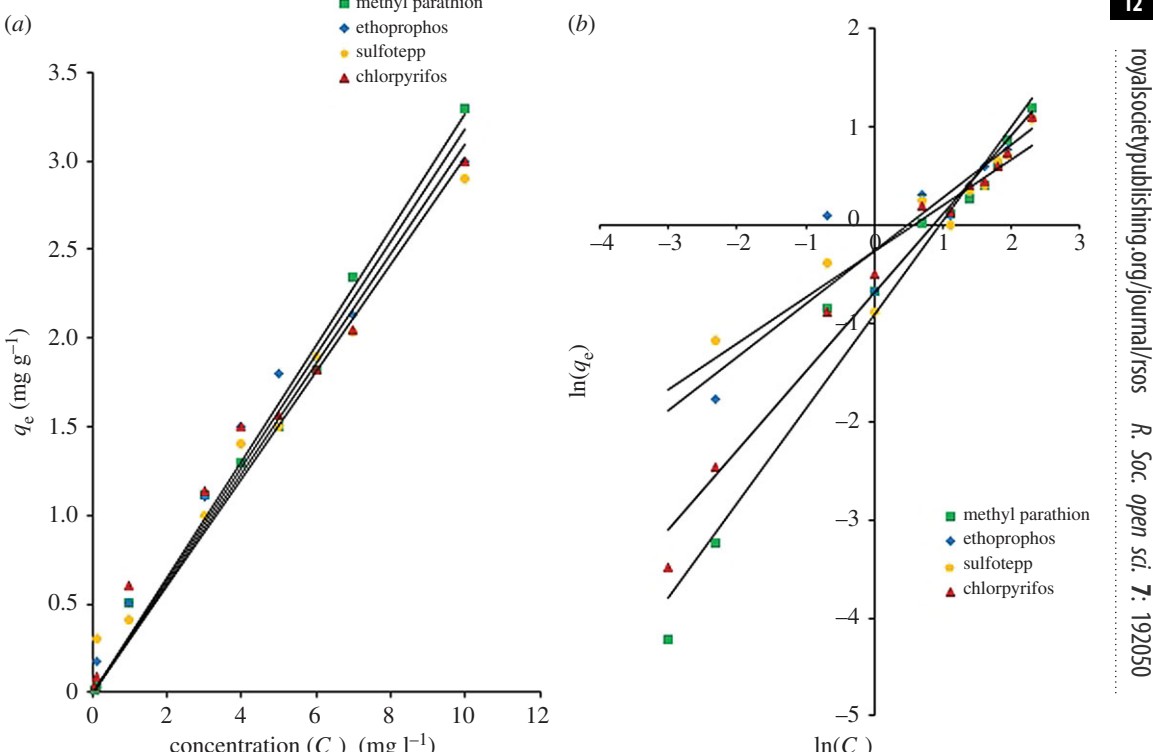

**Figure 7.** (*a*) Effect of concentration on SPMTA of OPPs and (*b*) Freundlich isotherm for the adsorption of OPPs onto GO/CNF–10 wt%. Adsorption conditions: 0.1 mg l$^{-1}$ of spiked OPPs in 10 ml sample solution; sample pH 12; adsorption time 15 min; 5 mg of GO/CNF–10 wt%.

first order rate constant ($k_1$). A very small value for the rate constant equates to a very slow reaction in general. Equally, a large value for the rate constant means a large value for the rate and the reaction is rapid.

These remarks are emphasized by the OPPs adsorption fitting onto GO/CNF adsorbent appropriated to the pseudo-second order kinetic model. The main rate-determining step suggested is chemisorption, in which the adsorption process involves chemical interactions by electron exchange or sharing between the analytes and the adsorbent [29]. Therefore, the results suggested that the predominant adsorption mechanism for this adsorbent system was the pseudo-second order.

## 3.5. Adsorbent dosage

To maximize the interactions between analytes and adsorption sites of the adsorbent, an adsorbent dosage is an essential factor, as it can acquire the quantitative adsorption of target analytes. Subsequently, the mass of the adsorbent GO/CNF–10 wt% was optimized for values between 5 and 30 mg using 0.1 mg l$^{-1}$ OPPs solutions. By logic, increasing the dosage of the adsorbent increases the adsorption capacity, as the active site of the adsorbent increases.

The results (figure 5*b*) presented that the adsorption of OPPs decreased with the increasing dosage of GO/CNF–10 wt% sorbent. This happened as the active sites for the adsorption are not saturated and the adsorption capacity is expressed by milligram adsorbed per gram of GO/CNF–10 wt%. From the graph, when increasing the dosage of the adsorbent up to 30 mg, the capacity of all the active sites on the surface of adsorbent is not fully used, thus leading to a decrease in adsorption per unit of adsorbent mass. Other than that, this may be attributed to slower or deficient adsorption from devices containing higher loading of adsorbent and overlap adsorbent making the adsorption sites overlapping and results in the increase of diffusion path length. The quantitative adsorption of OPPs was optimum using 5 mg of the GO/CNF–10 wt% sorbent. At 5 mg, the adsorption capacities were 0.829, 1.516, 0.859 and 0.904 mg g$^{-1}$ for methyl parathion, ethoprophos, sulfotepp and chlorpyrifos, respectively. Ethoprophos shows the highest adsorption followed by chlorpyrifos, sulfotepp and methyl parathion. The presence of S, N and O atoms in the structure assist in the higher adsorption of OPPs. Utilization of less than 5 mg of adsorbent in the SPMTA cannot be done, as the adsorbent was not well-fitted or lost from the pipette tip. Therefore, 5 mg of adsorbent was used for further analysis.

**Table 3.** Langmuir and Freundlich parameters for OPPs adsorption onto GO/CNF–10 wt%.

| | Langmuir isotherm | | | Freundlich isotherm | | |
|---|---|---|---|---|---|---|
| | $q_m$ (mg g$^{-1}$) | $K_L$ (l mg$^{-1}$) | $R^2$ | $K_F$ (mg g$^{-1}$) | $1/n$ | $R^2$ |
| methyl parathion | 9.20 | 0.05 | 0.243 | 0.40 | 0.96 | 0.966 |
| ethoprophos | 3.44 | 0.26 | 0.743 | 0.76 | 0.54 | 0.856 |
| sulfotepp | 3.42 | 0.22 | 0.655 | 0.76 | 0.47 | 0.863 |
| chlorpyrifos | 3.97 | 0.27 | 0.832 | 0.50 | 0.80 | 0.976 |

**Table 4.** Linear range, correlation coefficients ($R^2$), LOD and LOQ for the SPMTA of OPPs spiked in water samples ($n = 3$).

| | linearity range (mg l$^{-1}$) | $R^2$ | LOD (mg l$^{-1}$) | LOQ (mg l$^{-1}$) |
|---|---|---|---|---|
| methyl parathion | 0.05–10 | 0.998 | 0.01 | 0.03 |
| ethoprophos | 0.1–10 | 0.999 | 0.05 | 0.16 |
| sulfotepp | 0.1–10 | 0.998 | 0.05 | 0.16 |
| chlorpyrifos | 0.05–10 | 0.994 | 0.01 | 0.03 |

## 3.6. Analytes concentration and adsorption isotherm

Adsorption isotherms were used to provide an understanding of the interaction between sorbent and analytes and define the equilibrium relationship. It is also indicated in what way the adsorption molecules distribute between the solid and the liquid phase of sample solution when the adsorption process reaches an equilibrium state. The adsorbed amounts of OPPs (mg g$^{-1}$) were determined with the initial concentrations of OPPs and the results were shown in figure 7a. Five milligrams of GO/CNF–10 wt% were used in OPPs adsorption (0.05, 0.1, 0.5, 1, 2, 3, 4, 5, 6, 7 and 10 mg l$^{-1}$) within 15 min. The adsorption amounts were linearly increased with the OPPs initial concentration up to 10 mg l$^{-1}$. The calculated OPPs adsorptions were 0.015–3.30 mg g$^{-1}$ (methyl parathion), 0.017–3.001 mg g$^{-1}$ (ethoprophos), 0.305–2.90 mg g$^{-1}$ (sulfotepp) and 0.030–3.00 mg g$^{-1}$ (chlorpyrifos).

The equilibrium data were fitted to Langmuir and Freundlich isotherm models. The data fitted well in the Freundlich model (figure 7b) as all of the linear correlation coefficients ($R^2$) are greater than or equal to 0.856 compared to the Langmuir isotherm (greater than or equal to 0.243). From equation (2.5), $n$ (g l$^{-1}$) and $K_F$ (mg g$^{-1}$) are the Freundlich exponent and the Freundlich constant, respectively, and these related to the intensity of the adsorption process and were used to describe the pattern of the adsorption. The value of $1/n$ less than 1 is indicative of favourable adsorption. In truth, a negative value designates even stronger adsorption. The constant ($K$) indicates the affinity of sorbate towards the sorbent. $K_F$ (mg g$^{-1}$) has higher values than $K_L$ (l mg$^{-1}$). If $K$ value is larger, it implies that there is a strong interaction between OPPs analyte and the sorbent, whereas smaller value indicates a weak interaction. The Freundlich isotherm is based on the assumption that the adsorption process is a multilayer adsorption with no transmigration occuring between layers and the adsorption taking place on a heterogeneous surface on a multiple sites on the adsorbent medium. It assumes that the Langmuir isotherm applies to each layer and there is equal energy of adsorption for each layer except for the first layer [38]. This isotherm provides information about the heterogeneity of adsorption sites [39]. The ability of an analyte to bind is independent of whether nearby sites are occupied, which suggests that the binding process is bimolecular layer adsorption. The mechanism involves exchanging or sharing of electrons between the OPPs molecules and the active sites of the GO/CNF–10 wt%, known as valence-electron-driven adsorption [40]. That is why the Freundlich isotherm provides excellently fitting data of highly heterogeneous adsorbent systems. All parameters of isotherm constants ($K_F$, $1/n$ and correlation coefficients, $R^2$) and the maximum adsorption capacity ($q_m$) are summarized in table 3.

## 3.7. Desorption study

Desorption is an important factor in determining the reusability and regeneration ability of an adsorbent. Various desorbing agents were used in this study (1 M hydrochloric acid, methanol, toluene, dichloromethane and isopropyl alcohol). Based on that, only a few OPPs analytes (chlorpyrifos and

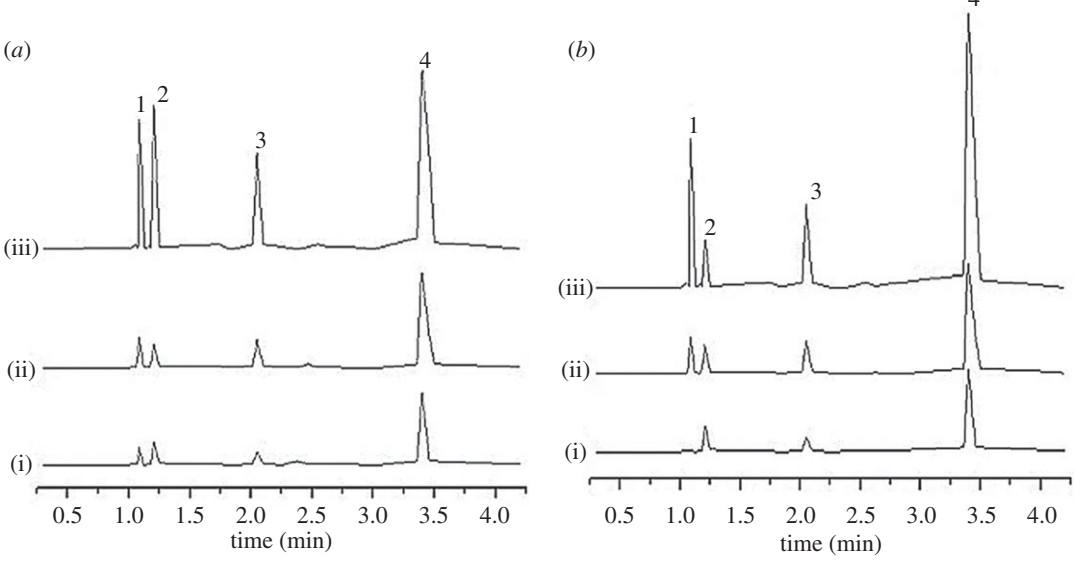

**Figure 8.** UPLC chromatograms of (*a*) cabbages and (*b*) rice before (i) and after (ii) spiking at 0.1 mg l$^{-1}$ and (iii) 0.5 mg l$^{-1}$. Peaks: 1, methyl parathion; 2, ethoprophos; 3, sulfotepp and 4, chlorpyrifos.

**Table 5.** Method precisions (RSD (%), $n = 3$) at two different concentrations for the SPMTA of OPPs in water samples.

| | intraday RSD (%) | | interday RSD (%) | |
|---|---|---|---|---|
| | 0.5 (mg l$^{-1}$) | 5 (mg l$^{-1}$) | 0.5 (mg l$^{-1}$) | 5 (mg l$^{-1}$) |
| methyl parathion | 0.06 | 2.02 | 0.17 | 0.87 |
| ethoprophos | 1.69 | 5.44 | 1.36 | 0.97 |
| sulfotepp | 2.34 | 2.31 | 7.76 | 1.38 |
| chlorpyrifos | 1.31 | 1.04 | 3.84 | 0.57 |

sulfotepp) shows some minimal desorption (%) with all desorbing agents. The desorption efficiencies of various desorption agents (1 M hydrochloric acid, methanol, toluene, dichloromethane and isopropyl alcohol) for chlorpyrifos and sulfotepp were 2.30%, 5.98%, 0.61%, 6.75%, 5.37% and 0%, 6.78%, 0.29%, 18.47%, 1.01%, respectively. No desorption was shown with other analytes (methyl parathion and ethoprophos).

From the inconsistent results, complete desorption was not possible at all because of the electrostatic reactions, and complexation occurred in which the adsorption process involves bimolecular layer adsorption and chemisorption interactions by electron exchange or sharing between the analytes and the adsorbent [39]. Apart from that, reusability of the adsorbent was impossible and the adsorbent can only be used in a single adsorption procedure in order to avoid any carry-over effect as based on the desorption results obtained, more than 70% of the analytes were still intact after the desorption. This desorption inability making the adsorption of OPPs analytes using the GO/CNF–10 wt% was acceptably well followed by the pseudo-second order model and the Freundlich isotherm as stated from the previous results.

## 3.8. Performance of the method

The performance of the developed SPMTA method was confirmed through the determination of repeatability, linearity, LOQ and LOD using the optimized conditions prior to real sample analysis. This SPMTA method was developed in order to tackle the trace concentration of OPPs analytes when applying to the real samples. The optimum parameters selected are as follows: sample pH 12, adsorption time of 15 min and adsorbent dosage 5 mg. In this study, the standards are added directly to the water sample solutions where a matrix-match calibration was constructed. Calibration curves of a matrix-matched were plotted for the 11 water samples spiked at 0.05–10 mg l$^{-1}$ for methyl parathion, ethoprophos, sulfotepp and chlorpyrifos. The LOD are those obtained with the UPLC–UV

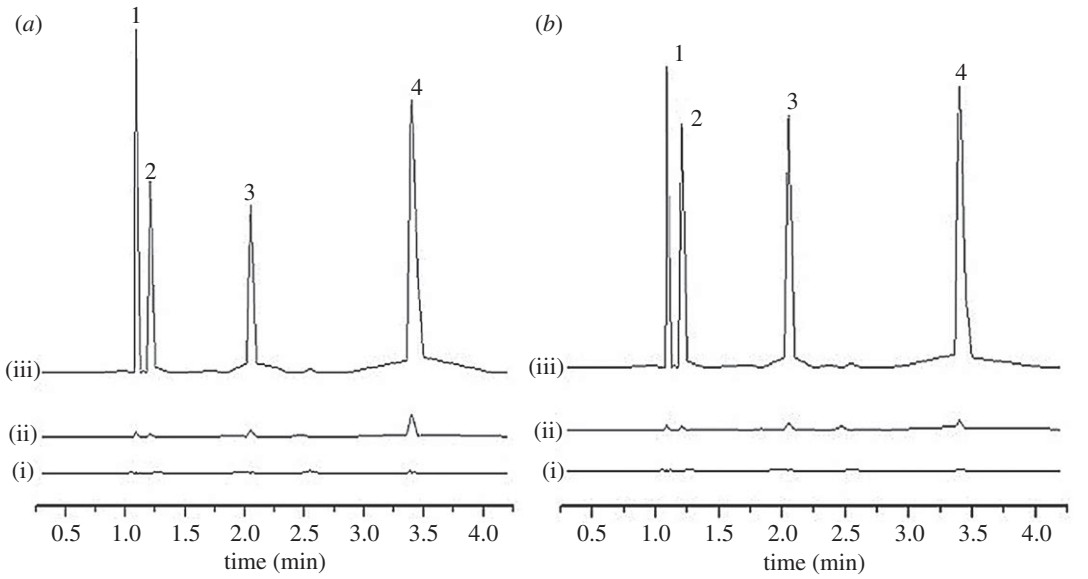

**Figure 9.** UPLC chromatograms of (*a*) Sungai Pahang and (*b*) Tasik Cheras before (i) and after (ii) spiking at 0.5 mg l$^{-1}$ and (iii) 5 mg l$^{-1}$. Peaks: 1, methyl parathion; 2, ethoprophos; 3, sulfotepp and 4, chlorpyrifos.

**Table 6.** Concentration of OPPs in non-spiked water and food samples. n.d., not detected.

| | Sungai Pahang | Tasik Cheras | cabbages | rice |
| --- | --- | --- | --- | --- |
| | concentration (mg l$^{-1}$) | concentration (mg l$^{-1}$) | concentration (mg l$^{-1}$) | concentration (mg l$^{-1}$) |
| methyl parathion | n.d. | n.d. | 0.32 | n.d. |
| ethoprophos | n.d. | n.d. | 2.45 | 2.52 |
| sulfotepp | n.d. | n.d. | 0.56 | 0.36 |
| chlorpyrifos | n.d. | n.d. | 0.88 | 1.71 |

**Table 7.** Consistency in adsorption (%) (*n* = 3) at two different concentrations for the SPMTA of OPPs in water and food samples.

| | Sungai Pahang | | Tasik Cheras | |
| --- | --- | --- | --- | --- |
| | consistency in adsorption (%) | | consistency in adsorption (%) | |
| | 0.5 (mg l$^{-1}$) | 5 (mg l$^{-1}$) | 0.5 (mg l$^{-1}$) | 5 (mg l$^{-1}$) |
| methyl parathion | 73.91 | 74.73 | 87.92 | 86.56 |
| ethoprophos | 82.05 | 87.95 | 84.94 | 79.85 |
| sulfotepp | 71.28 | 91.00 | 74.34 | 89.37 |
| chlorpyrifos | 80.58 | 88.81 | 74.45 | 72.17 |
| | cabbages | | rice | |
| | consistency in adsorption (%) | | consistency in adsorption (%) | |
| | 0.1 (mg l$^{-1}$) | 0.5 (mg l$^{-1}$) | 0.1 (mg l$^{-1}$) | 0.5 (mg l$^{-1}$) |
| methyl parathion | 76.47 | 82.68 | 84.89 | 90.18 |
| ethoprophos | 99.95 | 99.48 | 77.01 | 84.58 |
| sulfotepp | 85.11 | 71.14 | 71.41 | 76.03 |
| chlorpyrifos | 73.42 | 90.48 | 93.62 | 88.38 |

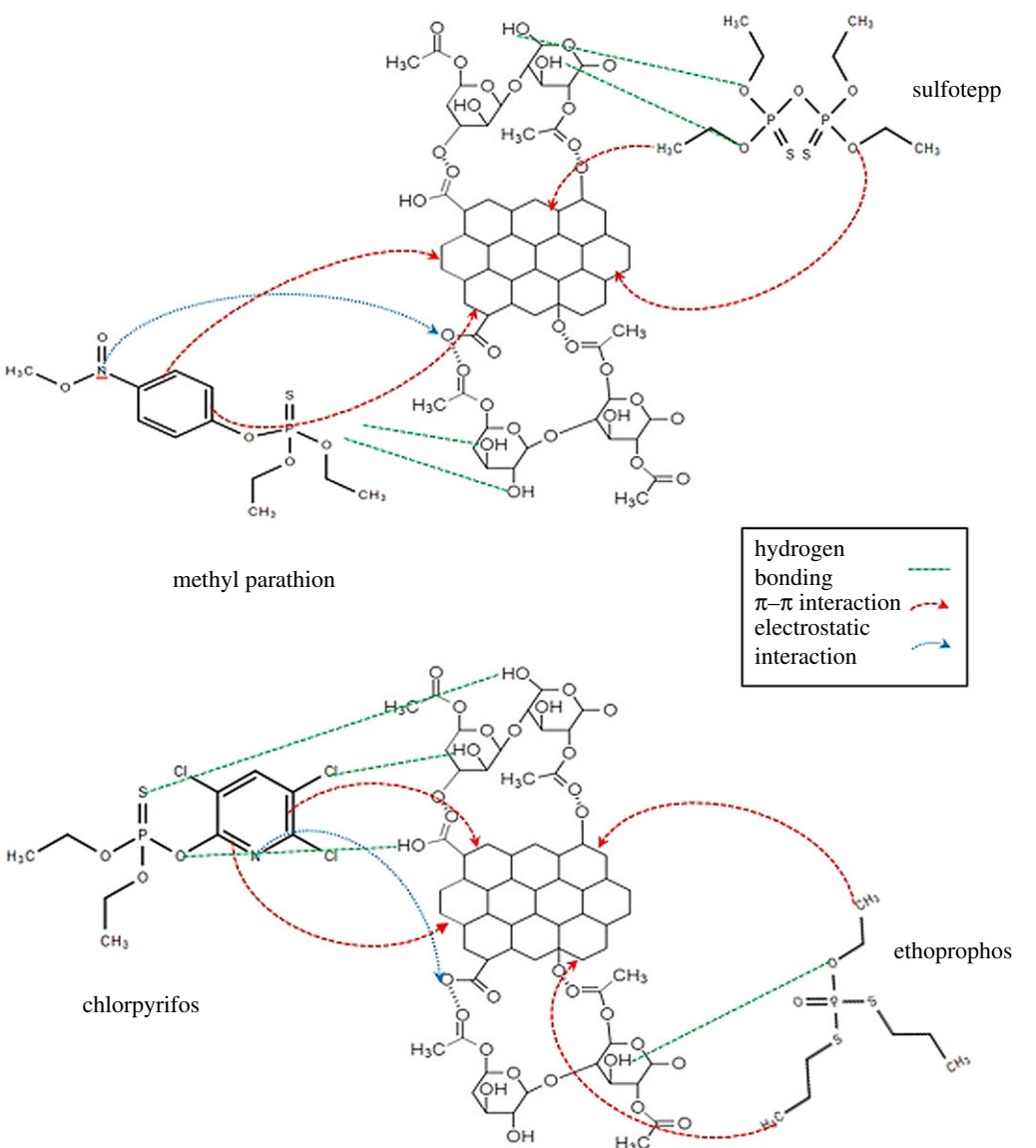

**Figure 10.** Proposed mechanism of interaction between OPPs analytes and GO/CNF–10 wt%.

method. Each analyte exhibited good linearity with determination coefficients ($R^2$) of 0.994, 0.998, 0.999 and 0.998 for chlorpyrifos, sulfotepp, ethoprophos and methyl parathion, respectively. The method also indicated low LOD for all analytes with the values of 0.01–0.05 mg l$^{-1}$ (table 4). These LOD values are valid for the adsorption of OPPs using the SPMTA method as the values obtained were found to be comparable to the previously reported method using HPLC–UV (0.01–0.09 mg l$^{-1}$) [27] and (0.01–0.03 mg l$^{-1}$) [41]. However, the LOD values are slightly higher than the reported MRLs set by the European Union and reported method using LC–UV (0.006–0.008 mg l$^{-1}$) [42].

The intra- and inter-day (RSDs) consistency in adsorption of the proposed method were determined with the OPPs spiked at two different concentrations (0.5 and 5 mg l$^{-1}$) in water samples (table 5). The intra- and inter-day precision for consistency in adsorption (RSDs) of OPPs were evaluated in between 0.06–5.44% ($n=3$) and 0.17–7.76% ($n=3$). The satisfactory results indicated that the developed SPMTA method successfully adsorbed the target analyte from water samples.

## 3.9. Application to water and food samples

The potential of the developed SPMTA method was investigated in the analysis of selected food and water samples, namely Sungai Pahang, Tasik Cheras, cabbages and rice. The samples were adsorbed

by using the optimized SPMTA procedure. The concentration of OPPs present in non-spiked food and water samples are summarized in table 6. It was found that all analytes were detected in the cabbages and only three analytes (ethoprophos, sulfotepp and chlorpyrifos) were detected in the rice sample (figure 8a and b). However, none of the analytes were detected in the water samples (Sungai Pahang and Tasik Cheras) (figure 9a and b). Thus, in determining the accuracy of the method, the samples were spiked at 0.5 and 5 mg l$^{-1}$ for water samples and 0.1 and 0.5 mg l$^{-1}$ for food samples, and the performances were evaluated. The percentage consistency in adsorption of the spiked samples are listed in table 7. Excellent consistency in adsorption was achieved for the spiked sample varied from 71.14 to 99.95%.

## 3.10. Superhydrophilic GO/CNF mechanism

Figure 10 shows the proposed interaction mechanism between the OPPs and the GO/CNF–10 wt% adsorbent. The interaction consists of three important features (electrostatic interactions, hydrogen bonding and π–π interactions). The electrospun nanofibre has hydroxyl groups, while the OPPs pesticides contain different anionic sites (N, O, S and Cl). Hydrogen in the adsorbent shows the binding abilities towards the anionic (N, O, S and Cl) groups. Consequently, the OPPs are adsorbed through strong hydrogen bonding. The GO/CNF–10 wt% structure has a variety of benzene ring, which contains a double bond (π–π electrons). Therefore, a variety of benzene rings and especially rich π–π electron arrangement makes the GO/CNF–10 wt% a suitable adsorbent for the adsorption of benzene-based pesticides [19].

Understanding how the analytes interact with the adsorbent is crucial in exploiting the differences among the four pesticides adsorbed by the GO/CNF–10 wt%. The abilities of electron donating of P, S, O and N atoms and the strong π-bonding network in the OPPs can generally assist adsorption. These OPPs have a different structure and structural formula. Thus, the π-bonding networks of all the OPPs are almost non-identical. Hence, the main causes of the differences might be because of the O, Cl and S atoms and the electrostatic interactions. First, the structures of OPPs are compared. All OPPs have an O atom and an S atom. Ethoprophos and sulfotepp, which contain two S atoms have a higher ability to donate electrons, as the electron-donating abilities assisted the adsorption. The adsorption capacity is also better than that of methyl parathion and chlorpyrifos based on the donating electron ability of S, N and O atoms. Methyl parathion has an additional branched structure that weakens the electrostatic type interaction and π-bonding network of benzene in its structure. That is why methyl parathion shows the worst performance among the four OPPs. This branched structure is not helpful to the interactions between methyl parathion and the GO/CNF–10 wt%. Second, chlorpyrifos contains chlorine atoms, which have an electron-withdrawing nature. Chlorine atoms could lead to an acidic π system. It forms complexation with the π system with no derivative and benefits the adsorption process [19]. Thus, it can be concluded that the electron-donating abilities of P, S and N atoms and the strong π-bonding network of benzene promoted the adsorption.

# 4. Conclusion

In conclusion, the formation of GO/CNF was confirmed by FTIR analysis, and the spectra of the GO-incorporated CA hybrid nanofibres revealed the presence of functional groups whose absorption frequencies correspond to –OH, C=O, C–CH$_3$ and C–O stretching and bending. Superhydrophilic GO/CNF–10 wt% was chosen as an adsorbent based on the adsorbent diameter and the preliminary adsorption of OPPs. The specific surface area of the GO/CNF–10 wt% from BET was 3.391 m$^2$ g$^{-1}$. Adsorption results indicated that higher adsorption capacity was influenced by the higher initial pH value. The optimum adsorption conditions of pH 12, 5 mg of adsorbent and 15 min of adsorption time were selected as they gave the highest adsorption capacity for all OPPs. The adsorption data fit the Freundlich adsorption model and well followed the second-order kinetic model. Acceptable precision (RSDs) was achieved for intraday (0.06–5.44%, $n = 3$) and interday (0.17–7.76%, $n = 3$) analyses and excellent consistency in adsorption were achieved ranging from 71.14 to 99.95%, which were obtained for the spiked OPPS from river and lake water samples as well as food samples. The LOD obtained for all analytes were in the range of 0.01–0.05 mg l$^{-1}$ and the LOQ were in between 0.03 and 0.16 mg l$^{-1}$. The results of the synthesized GO/CNF–10 wt% indicate its inexpensive and efficient adsorption of OPPs (methyl parathion, ethoprophos, sulfotepp and chlorpyrifos) in water and food samples with no extra peak in the UPLC chromatogram indicating the selective nature of the

SPMTA method. Additionally, excellent stability and the low cost of GO/CNF–10 wt% make them the most promising materials for the analytes adsorption in food and water samples. As mentioned above, with the low cost of the adsorbent, the cost of the adsorption method through the SPMTA technique also reduced as compared with other methods. Therefore, the newly synthesized GO/CNF–10 wt% showed high potential as an adsorbent for OPPs analysis.

Data accessibility. Data have been uploaded as part of the electronic supplementary material.

Author's contribution. N.I.F.A. carried out the most of the laboratory work, accomplished the data analysis and drafted the manuscript. S.K. participated in the design of the study, data analysis and finalized the manuscript. N.A.R., N.Y., M.H.W. and A.S.A.K. coordinated the study and helped to draft the manuscript. All authors gave approval for publication.

Competing interests. All authors declare that there is no conflict of interest.

Funding. This work was supported by the Universiti Putra Malaysia (grant no. 9643100) and the Ministry of Higher Education, Malaysia for Fundamental Research Grant Scheme (grant no. 5540122).

Acknowledgements. The authors would like to thank officers from the Department of Chemistry, Faculty of Science and Faculty of Food Science and Technology Universiti Putra Malaysia for the guidance and advice towards the research.

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
