## [Reviewer comments · Royal Society Open Science]

Review History

RSOS-192050.R0 (Original submission)

Review form: Reviewer 1

Is the manuscript scientifically sound in its present form?

Yes

Are the interpretations and conclusions justified by the results?

Yes

Is the language acceptable?

Yes

Do you have any ethical concerns with this paper?

No

Have you any concerns about statistical analyses in this paper?

No

Recommendation?

Major revision is needed (please make suggestions in comments)

Comments to the Author(s)

The manuscript describes a GOOD work AND is well presented. Authors need following points to be included before reconsideration for possible publication:

1. The introduction section is very week. Some information about importance of this research and application of adsorbents as well as removal of Pesticides should add to this section. The following references suggested for upgrading this section and should be cited:

- Iranian Journal of Chemistry and Chemical Engineering, 2017; 36(6), pp. 127-137.

- Desalination and Water Treatment; 2017; 80, pp. 247-254.

- Iranian Journal of Chemistry and Chemical Engineering, 2017; 36(3), pp. 25-36

- Desalination and Water Treatment, 2017, 100, pp. 116-125.

- Desalination and Water Treatment, 70, pp. 290-293.

2. The exact information about the apparatus used in this study should be presented in materials and method section.

3. The methods are not sufficiently informative of how the experimental study was carried out. the adsorption experiment should exactly explained.

4. In some places, there is a lack of basic theoretical analysis, but only a simple statement of the experimental results. The causes and mechanisms of these phenomena should be analyzed at a deeper level.

5. For interpretation of the results of isotherm and kinetic studies, The following references suggested for use and cite:

- Desalination and Water Treatment, 2017, 70, pp. 355-363.

- Environment Protection Engineering, 2016,42(1), pp. 149-168.

- Fluoride: 50(2), pp. 256-268

6. What about the cost of this method to adsorption compared with the other methods?

Review form: Reviewer 2

Is the manuscript scientifically sound in its present form?

Yes

Are the interpretations and conclusions justified by the results?

Yes

Is the language acceptable?

Yes

Do you have any ethical concerns with this paper?

No

Have you any concerns about statistical analyses in this paper?

No

Recommendation?

Accept with minor revision (please list in comments)

Comments to the Author(s)

This is a well-written paper containing interesting results which merit publication. However, I found a terrible mistake in Fig. 5 and 6. Assuming all OPPs were adsorbed, the maximum adsorption capacity should be 0.2 mg/g, if 5 mg adsorbents were used. But, this is not the case in Fig. 5 and 6a. The authors are required to recheck their adsorption data and provide a reasonable

explanation or clarification. In addition, all the origin graphs are vague in the manuscript, and thus they should be replaced with distinct ones without ambiguity.

Decision letter (RSOS-192050.R0)

13-Jan-2020

Dear Dr Kamaruzaman:

Title: Superhydrophilic Graphene Oxide/Electrospun Cellulose Nanofiber for Efficient Adsorption of Organophosphorus Pesticides from Environmental Samples
Manuscript ID: RSOS-192050

The editor assigned to your manuscript has now received comments from reviewers. We would like you to revise your paper in accordance with the referee and Subject Editor suggestions which can be found below (not including confidential reports to the Editor). Please note this decision does not guarantee eventual acceptance.

Please submit your revised paper before 05-Feb-2020. Please note that the revision deadline will expire at 00.00am on this date. If we do not hear from you within this time then it will be assumed that the paper has been withdrawn. In exceptional circumstances, extensions may be possible if agreed with the Editorial Office in advance. We do not allow multiple rounds of revision so we urge you to make every effort to fully address all of the comments at this stage. If deemed necessary by the Editors, your manuscript will be sent back to one or more of the original reviewers for assessment. If the original reviewers are not available we may invite new reviewers.

Royal Society of Chemistry
Thomas Graham House
Science Park, Milton Road
Cambridge, CB4 0WF

Royal Society Open Science - Chemistry Editorial Office

RSC Associate Editor:
Comments to the Author:
(There are no comments.)

RSC Subject Editor:
Comments to the Author:
(There are no comments.)

Reviewers' Comments to Author:
Reviewer: 1

Comments to the Author(s)

The manuscript describes a GOOD work AND is well presented. Authors need following points to be included before reconsideration for possible publication:

1. The introduction section is very week. Some information about importance of this research and application of adsorbents as well as removal of Pesticides should add to this section. The following references suggested for upgrading this section and should be cited:

- Iranian Journal of Chemistry and Chemical Engineering, 2017; 36(6), pp. 127-137.
- Desalination and Water Treatment; 2017: 80, pp. 247-254.
- Iranian Journal of Chemistry and Chemical Engineering, 2017; 36(3), pp. 25-36
- Desalination and Water Treatment, 2017, 100, pp. 116-125.
- Desalination and Water Treatment, 70, pp. 290-293.

2. The exact information about the apparatus used in this study should be presented in materials and method section.

3. The methods are not sufficiently informative of how the experimental study was carried out. the adsorption experiment should exactly explained.

4. In some places, there is a lack of basic theoretical analysis, but only a simple statement of the experimental results. The causes and mechanisms of these phenomena should be analyzed at a deeper level.

5. For interpretation of the results of isotherm and kinetic studies, The following references suggested for use and cite:

- Desalination and Water Treatment, 2017, 70, pp. 355-363.
- Environment Protection Engineering, 2016,42(1), pp. 149-168.
- Fluoride: 50(2), pp. 256-268

6. What about the cost of this method to adsorption compared with the other methods?

Reviewer: 2

Comments to the Author(s)

This is a well-written paper containing interesting results which merit publication. However, I found a terrible mistake in Fig. 5 and 6. Assuming all OPPs were adsorbed, the maximum adsorption capacity should be 0.2 mg/g, if 5 mg adsorbents were used. But, this is not the case in Fig. 5 and 6a. The authors are required to recheck their adsorption data and provide a reasonable explanation or clarification. In addition, all the origin graphs are vague in the manuscript, and thus they should be replaced with distinct ones without ambiguity.

Author's Response to Decision Letter for (RSOS-192050.R0)

See Appendix A.

Decision letter (RSOS-192050.R1)

11-Feb-2020

Dear Dr Kamaruzaman:

Title: Superhydrophilic Graphene Oxide/Electrospun Cellulose Nanofiber for Efficient Adsorption of Organophosphorus Pesticides from Environmental Samples
Manuscript ID: RSOS-192050.R1

It is a pleasure to accept your manuscript in its current form for publication in Royal Society Open Science. The chemistry content of Royal Society Open Science is published in collaboration with the Royal Society of Chemistry.

RSC Associate Editor
Comments to the Author:
(There are no comments.)

Reviewer(s)' Comments to Author:

Appendix A

NO.	COMMENTS	CORRECTIONS MADE	PAGE NO.
1	The introduction section is very weak. Some information about importance of this research and application of adsorbents as well as removal of Pesticides should add to this section. The following references suggested for upgrading this section and should be cited.	Organophosphorus pesticides (OPPs) are among the most broadly used pesticides with high toxicity in the world since the 1970s. Their popularity is largely contributed by their favorable characteristics such as short persistence and biodegradable Over so many years, pesticides been used by growers and farmers for food production in order to increase the quality and the production of crops and against a wide range of pests. In a consequence, consumers are expose to pesticides even though the exposure are usually in a small number of quantities. OPPs may have effects on untargeted organisms including humans upon a frequent exposure for a long period of time to even low (ppb) levels of OPPs from air, water and food due to the acetylcholinesterase inhibition in the body and reported to exhibit teratogenicity, cytotoxicity, immunotoxicity and genotoxicity [2,6]. Thus result in dysfunction of many behavioral and autonomic systems, eventually leading to muscles and respiratory weakness or paralysis as well as death. Contamination of agricultural products and environmental water has become a major concern, as OPPs are one of the most common classes of pesticides involved in poisoning. Stability and difficult degradation of OPPs are due to complex aromatic structure of them. Thus, analysis and monitoring of trace levels of OPPs in environmental water and food contamination are very crucial as for environmental control.	1, 2, 3 & 4

OPPs determination constitutes a real analytical challenge as they are recognizable in very low concentration levels of environmental water and food samples. Since the OPPs concentration in the real samples are in trace amounts, various techniques have been developed like photocatalysis, oxidation, coagulation and flocculation, ion-exchange and adsorption for the adsorption of these pesticides. Adsorption of these pesticides by using oxidation method is difficult as this method has significant limitations, which in the massive production of iron sludge and high cost requirements. Coagulation and flocculation of pesticides produces large amounts of sludge at their disposal which becomes a problem. Adsorption has recently considered as a capable, acceptable and most important technique for decreasing concentration of OPPs from aqueous solutions. Recently, efforts have been made to use more affordable adsorbent in the adsorption of pesticides.

Various studies have been done by using CA as an adsorbent in an adsorption of different contaminants such as, separation/preconcentration of silver(I) and lead(II) in environmental samples on cellulose nitrate membrane filter prior to their flame atomic absorption spectrometric determinations, cellulose cellets as new type of adsorbent for the removal of dyes from aqueous media and preparation of cellulose/graphene composite and its applications for triazine pesticides adsorption from water. CA is highly hydrophilic in nature as it consist of high density of carbonyl and hydroxyl groups

However, CA have some limitations in in terms of high moisture absorption, associated dimensional instability as well

		as ultraviolet (UV) stability. CA also can be easily attacked by concentrated alkalies and acids. The effect of initial pH, contact time, adsorbent dosage, initial concentration of OPPs and desorption solvent was evaluated. In addition, adsorption kinetics and isotherm study were also investigated and analyzed.	
2	The exact information about the apparatus used in this study should be presented in materials and method section.	Graphene oxide/electrospun cellulose nanofiber (GO/CNF) was synthesized using electrospinning machine through syringe pump (NE-300 Just Infusion™ Syringe Pump) which was connected through a high voltage power supply (RXN-305D, China) at 2 mL/h of flow rate by 18 gauge needle. Fourier Transform Infrared (FTIR) spectra (4000-500 cm⁻¹) were recorded on a Perkin Elmer (1600 series) spectrometer (Waltham, MA, US) using Attenuated Total Reflectance (ATR) technique. The morphology, structure and diameter of the newly GO/CNF were observed using a JEOL JSM-6700F field emission scanning electron microscopy (FESEM) (Tokyo, Japan) operated at 5.0 kV, magnification 10000× and a working distance of 4.3 mm. OPPs analysis were carried out using UPLC through a Supelco Ascentis® C18 column (15 cm × 4.6 mm, 5 μm) (Waters, Ireland).	4
3	The methods are not sufficiently informative of how the experimental study was carried out. the adsorption experiment should exactly explained.	Each parameters were optimized accordingly in a sequence, pH, time, adsorbent dosage, initial concentration and desorption study. Followed by performance of the method and application to the real samples.	7
4	In some places, there is a lack of basic theoretical analysis, but only a simple statement of the experimental results. The	Sample pH	

causes and mechanisms of these phenomena should be analyzed at a deeper level.	The distribution of the negative and positive charges may have huge effects on an adsorption by the adsorbent. Thus, the balance between them is one of the most important environmental factors that affects the adsorbents potential. From the graph, the highest adsorption was achieved at pH 12 with adsorption capacity of 0.25, 0.43, 0.34 and 0.23 mg/g for methyl parathion, ethoprophos, sulfotepp and chlorpyrifos respectively. Ethoprophos shows the highest adsorption among all analytes while methyl parathion was the worst. Ethoprophos has higher donating electron ability of S, N, and O atoms as compared to other analytes. Two S atoms are present in the structure, which assisted with the higher adsorption. Other than that, at this point, OPPs analytes undergo hydrolysis rather than adsorption [19]. Hydrolysis is due to the reaction with water. At higher pH, water reacts with alkaline solution and the analyte compounds itself broken down by alkaline environment. Thus making a significant contribution to the OPPs analytes removal. Adsorbent Dosage By logic, increasing the dosage of the adsorbent increases the adsorption capacity as the active site of the adsorbent increases. This happened as the active sites for the adsorption are not saturated and the adsorption capacity is expressed by milligram adsorbed per gram of GO/CNF–10 wt%. From the graph, when increasing the dosage of the adsorbent up to 30 mg, the capacity of all the active sites on the surface of adsorbent is not fully utilized, thus leading to a decrease in adsorption per unit of adsorbent mass.	11, 12, 14 & 16
---	---	-----------------------------------

		At 5 mg, the adsorption capacity were 0.829, 1.516, 0.859 and 0.904 mg/g for methyl parathion, ethoprophos, sulfotepp and chlorpyrifos respectively. Ethoprophos shows the highest adsorption followed by chlorpyrifos, sulfotepp and methyl parathion. The present of S, N and O atoms in the structure assist in the higher adsorption of OPPs. Desorption Study Desorption is an important factors in determining the reusability and regeneration ability of an adsorbent.	
5	For interpretation of the results of isotherm and kinetic studies, The following references suggested for use and cite	Contact Time and Adsorption Kinetic Kinetic studies, in which the relationship between the contact time and adsorption capacity are among the most important studies related to adsorption. The speed of the adsorption process can be predicted and the data from these predictions can be used for the modeling and designing of an adsorption process. At 15 minutes, the adsorption capacity of GO/CNF–10 wt% for 0.1 mg/L of methyl parathion, ethoprophos, sulfotepp and chlorpyrifos were 0.228, 0.421, 0.243 and 0.258 mg/g respectively. Methyl parathion shows the lowest adsorption capacity, as an additional branched in the structure weakens the π bonding network of benzene in its structure and the electrostatic type interaction. This branched structure is not conducive to the interactions between the methyl parathion and the fiber itself. The rate constant, k, gives a direct measure of the relative reaction rate. Pseudo second order rate constant (k_2) has a higher values compared to pseudo first	12, 13 & 15

		order rate constant (k_1). A very small value for the rate constant equates to a very slow reaction in general. Equally, a large value for the rate constant means a large value for the rate and the reaction is rapid. Analytes Concentration and Adsorption Isotherm Freundlich isotherm is based on the assumption that the adsorption process is a multilayer adsorption with no transmigration occurs between layers and the adsorption takes place on a heterogeneous surface on a multiple sites on the adsorbent medium. It assumes that the Langmuir isotherm applies to each layer and there is equal energy of adsorption for each layer except for the first layer.	
6	What about the cost of this method to adsorption compared with the other methods?	As mentioned above, with the low cost of the adsorbent, the cost of the adsorption method through SPMTA technique also reduced as compared with other methods.	19
7	However, I found a terrible mistake in Fig. 5 and 6. Assuming all OPPs were adsorbed, the maximum adsorption capacity should be 0.2 mg/g, if 5 mg adsorbents were used. But, this is not the case in Fig. 5 and 6a. The authors are required to recheck their adsorption data and provide a reasonable explanation or clarification. In addition, all the origin graphs are vague in the manuscript, and thus they should be replaced with distinct ones without ambiguity.	Each parameter has different adsorption condition and each of them was perform in a sequence while keeping others in constant. pH: 0.1 mg/L of spiked OPPs in 10 mL sample solution; adsorption time 20 min; 15 mg of GO/CNF-10 wt% time: 0.1 mg/L of spiked OPPs in 10 mL sample solution; sample pH 12; 15 mg of GO/CNF-10 wt% adsorbent dosage: 0.1 mg/L of spiked OPPs in 10 mL sample solution; adsorption time 15 min; sample pH 12 analytes concentration: 0.1 mg/L of spiked OPPs in 10 mL sample solution;	

		sample pH 12; adsorption time 15 min; 5 mg of GO/CNF-10 wt%	
--	--	---	--